# Welfare and Performance of Post-Weaning Sows and Piglets Previously Housed in Farrowing Pens with Temporary Crating on a Spanish Commercial Farm: A Pilot Study

**DOI:** 10.3390/ani12060724

**Published:** 2022-03-13

**Authors:** Heng-Lun Ko, Déborah Temple, Janni Hales, Xavier Manteca, Pol Llonch

**Affiliations:** 1Department of Animal and Food Science, School of Veterinary Science, Universitat Autònoma de Barcelona, Bellaterra, 08193 Barcelona, Spain; xavier.manteca@uab.cat (X.M.); pol.llonch@uab.cat (P.L.); 2AWEC Advisors SL, Parc de Recerca de la Universitat Autònoma de Barcelona, Bellaterra, 08193 Barcelona, Spain; deborah.temple@uab.cat; 3SKIOLD A/S, La Cours Vej 1, 7430 Ikast, Denmark; jhp@skiold.com

**Keywords:** aggression, behavior, farrowing system, pig, stress biomarker, temporary crating, vocalization, weaning

## Abstract

**Simple Summary:**

Social pressure, especially in Europe, towards the ban of farrowing crates has driven research on alternative farrowing systems. However, few studies comparing the welfare and performance of pigs in different types of farrowing pens with temporary crating have been carried out under commercial conditions. Additionally, the weaning stress response of pigs that are previously housed in alternative farrowing systems is overlooked. Therefore, the present study aimed to compare how the sows and piglets from three types of farrowing systems adapt to the weaning challenge. Behavioral observation, number of skin lesions, and salivary stress biomarkers were carried out at different time points around weaning, to assess the adaptability of sows and piglets to weaning. Our results found that the post-weaning aggression level in piglets from three farrowing systems was similar. Piglets’ saliva samples from one of the alternative farrowing systems showed a reduced stress response at weaning, which may suggest better weaning adaptability of these pigs. Sows from the same alternative farrowing system vocalized most on the day of weaning and decreased drastically over time. This vocalization frequency pattern after weaning might indicate a greater effect of abrupt separation from the piglets in this type of farrowing system.

**Abstract:**

The study investigated the effect of farrowing environment on the weaning adaptability of sows and piglets. One farrowing crate (FC) and two farrowing pens with temporary crating (TC: SWAP and JLF15) were compared. Sixty-four sows and 663 piglets were followed until 5 days post-weaning. At weaning (D24), sows and piglets were moved to group pens and nursery pens, respectively. Sows and piglets’ behaviors were observed on D24, D25, and D26. On D23, D25, and D26, piglets’ skin lesions were counted, and sows and piglets’ saliva samples were collected for stress biomarkers (cortisol and chromogranin A, CgA). Piglets were weighed on D23 and D29. All the piglets’ skin lesions increased on D25 and decreased on D26 (*p* < 0.05). Compared to D23, cortisol of JLF15 and CgA of FC piglets increased, whereas those of SWAP piglets remained similar after weaning (*p* < 0.05). Post-weaning performance in piglets was similar across farrowing systems. SWAP sows vocalized more than FC and JLF15 on D24 and D25 (*p* < 0.001). Results suggested that SWAP piglets showed a lower weaning stress response. Frequent post-weaning vocalization in SWAP sows might be linked with a negative effect of the abrupt separation from the piglets.

## 1. Introduction

In modern pig farming, farrowing crates are used to reduce the risk of piglet mortality caused by sow crushing [1]. Currently, approximately 95% of the sows are housed in the farrowing crates during farrowing and lactation in the European Union (EU) [2]. However, farrowing crates jeopardize the welfare of the sows [3]. Farrowing crates limit the behavioral repertoire of body movements and restrict the expression of natural behaviors like nest-building and mother–young interactions [3]. Along with the growing societal trend of eliminating farrowing crate policies in several EU countries, and the well-received European citizen initiative “End the Cage Age” launched in 2018, there are many alternative farrowing systems being studied and implemented lately. A farrowing pen with temporary crating (TC) is one of them. It allows the producers to crate the sows temporarily for a few days when the piglets are newly born and vulnerable, to prevent piglet crushing, and to set loose the sows for the rest of the lactation period, to ensure freedom of body movements of the sows.

While there is increasing research on welfare assessment of the alternative farrowing systems in sows and piglets during lactation, few studies followed these animals after weaning. Weaning in pig production is usually performed abruptly which makes it one of the most stressful events in a pig’s life on commercial farms [4]. Yet, several studies have found benefits of the enriched early environment to mitigate weaning stress in piglets [5], including pre-weaning socialization [6,7,8] and environmental enrichment [8,9,10]. However, little research has focused on how farrowing systems that facilitate the mother–young interactions play a role in coping with weaning stress. TC has shown to facilitate the mother–young interactions [1,11], and as Telkänranta and Edwards [12] stated, a sow is “the most important social figure in the early life of a pig” for piglets, which underlines the importance of the social experiences with the sow during suckling.

This paper is a follow-up study of Ko et al. [11], in which we compared the welfare and performance of sows and piglets in three farrowing systems: one conventional system with a farrowing crate (FC) and two TCs. This paper focuses on the same group of sows and piglets in post-weaning, until 5 days (D) after weaning.

We hypothesized that piglets raised in an early environment that allows more mother–young interactions would better cope with abrupt weaning stress, due to previously learned social skills from the mother [12], heavier body weight [13], larger farrowing pen size to play in, and the nesting material to manipulate with [9,14] (and possibly earlier exposure/consumption of solid food [15], i.e., the nesting material). We also hypothesized that sows that can interact with the piglets during lactation, would have more stress after abrupt weaning due to a stronger bond with the piglets [16]. The aim of the study was to study the effect of farrowing systems on behavior, aggression-related skin lesions, salivary stress, and performance in post-weaning sows and piglets from FCs, compared to two different TCs.

## 2. Materials and Methods

### 2.1. Housing and Experimental Design

The study took place on a farrow-to-finish commercial farm in Girona, Spain. During lactation, three farrowing systems were used, one FC and two TC, including Sow Welfare and Piglet protection pen (SWAP) and JLF15 (both produced by SKIOLD A/S, Ikast, Denmark). In each batch, there were five FC pens, six SWAP pens, and six JLF15 pens during lactation. Five FC pens were installed in one room, and six SWAP pens and six JLF15 pens were installed in another room. Sows entered the rooms a week before the expected farrowing date. FC sows were confined from entry to weaning, whereas TC sows were confined from 1D before expected farrowing to 3D after farrowing. Four batches of crossbred Duroc sows (*n* = 64; 18, 23 and 23 in FC, SWAP and JLF15, respectively) with a total of 663 piglets (183, 243 and 237 in FC, SWAP and JLF15, respectively) were followed in the study from lactation to 5D after weaning in four seasons between 2018 and 2019: autumn batch (November), winter batch (February), spring batch (May) and summer batch (July). Regarding the technical details (dimension of the pens, crates, and creep areas) of the three farrowing systems used in the study, see Figure 1 and Table 1 in Ko et al. [11].

Weaning occurred in the morning on D24. Sows and piglets of the four batches were removed from the farrowing pens and were moved to 14 group pens (5, 3, 3 and 3 pens in each batch) and 16 nursery pens (7, 3, 3 and 3 pens in each batch), respectively. Pigs were randomly regrouped but were kept with those from the same farrowing system together. Pigs that were regrouped with those from different farrowing systems after weaning, were considered as surplus sows/piglets and were removed from the study. Due to different availabilities of the pens for sows and piglets, and the production cycle of the sows (e.g., some were culled or were served as nurse sows), the number of pigs per pen after weaning was not consistent across farrowing systems and across the four batches (as specified in Figure 1a,b), making it 56 sows (16, 21 and 19 in FC, SWAP and JLF15) and 491 piglets (155, 168 and 168 in FC, SWAP and JLF15) in the end. The group pen size for the weaning sows was 2.2 m × 3.5 m (four sows/pen on average) and was installed with one feeder and one drinker. The nursery pens of the autumn batch were 2.4 m × 2.53 m (36.7 piglets/pen on average) (installed with two feeders and three drinkers per pen) and 1.2 m × 2.55 m (16.5 piglets/pen on average) (installed with one feeder and two drinkers per pen). The nursery pens of the rest of the batches were 2.4 m × 2.8 m (35 piglets/pen on average) (installed with two feeders and four drinkers per pen).

### 2.2. Animals and Management

Sows in the group pens and piglets in the nursery pens had ad libitum feed and water access. Sows had home-mixed feed which was barley- and soybean-based. Piglets had commercial nursery feed (Nuscience, Ghent, Belgium) and were supplemented with complementary liquid feed (Re-hydralab; Labiana Life Sciences S.A., Terrassa, Spain) during the first 5 days after weaning.

The group pens for the sows after weaning were in a semi-open building which was dependent on natural lighting and ventilation. The nursery for the weaning piglets was an indoor building, in which light was on from 07:00 to 18:00 and the room temperature was programmed at 25 ± 2 °C.

The study of the sows ended 2D after weaning, and the piglets ended 5D after weaning. Sows and piglets followed their production cycle after the study. On the 3rd and 4th days after weaning, sows were artificially inseminated and were moved to a larger pen with a dynamic group of gestating sows altogether. On the 6th day after weaning, the nursery feed for piglets changed.

### 2.3. Direct Behavioral Observation in Sows and Piglets after Weaning

Direct behavioral observations were conducted on D24 (the day of weaning), D25 (1D post-weaning), and D26 (2D post-weaning) by one observer. It was conducted hourly between 14:00 and 17:00 on each observation day, making it three sessions per day (30-s scan-sampling for 3 min per pen). Each session started observing the sows in the group pens and then the piglets in the nursery pens. Behavior categories for the piglets and the sows after weaning are listed in Table 1 and Table 2, respectively.

### 2.4. Pre-Selection of the Piglets for Lesion Scoring and Saliva Collection

Of each litter, six piglets were selected: the heaviest, the middle, and the lightest on D3 (3 days after farrowing) of males and females. To assess the intensity of aggression in piglets after weaning, and to study the effect of weaning stress in sows and piglets, three sampling days were determined: D23 (1D pre-weaning), D25 (1D post-weaning) and D26 (2D post-weaning).

#### 2.4.1. Skin Lesion Scoring in Piglets around Weaning

The number of skin lesions was scored as suggested by Turner et al. [20] by one observer. The piglet’s body was divided into six parts: the left and the right sides of the front (i.e., from head to front leg), middle (i.e., the trunk), and rear (i.e., hind leg to rump) parts of the body. In each body part, only fresh and unbroken linear lesions were scored.

#### 2.4.2. Sows and Piglets’ Saliva Collection around Weaning and Stress Biomarker Analysis

Saliva samples were collected to determine two salivary stress biomarkers, cortisol (CORT) and chromogranin A (CgA). Saliva samples were collected with the swab in the Salivette^®^ tube (Sarstedt, Aktiengesellschaft & Co., Nümbrecht, Germany) to the pig’s mouth for 1 min. Sow samples were collected between 09:00 and 10:00, and piglet samples between 10:30 and 14:00. Samples were centrifuged (Heraeus™ Labofuge™ 200 Centrifuge; Thermo Fisher Scientific GmbH, Dreieich, Germany) for 10 min at 3000 rpm and stored at −20 °C until analysis. CORT was detected by the automated chemiluminescence immunoassay (Immulite 1000 cortisol; Siemens Medical Solutions Diagnostics, Los Angeles, CA, USA) and CgA by the time-resolved immunofluorometry assays (TR-IFMA).

### 2.5. Weighing of Piglets

All the piglets were weighed. To calculate the average daily gain (ADG) of each individual piglet, two weighing points were set: one on D23 (1D pre-weaning) and the other one on D29 (5D post-weaning). Bodyweight on D3 (3 days after birth, i.e., the initial weight after litter stabilization) was obtained. Weighing on D23 was carried out after saliva sampling to avoid the confounding effect of handling stress.

### 2.6. Statistical Analysis

Data were analyzed in RStudio version 1.2.5033 (R Foundation, Vienna, Austria). The experimental unit for all the analyses is the pen. Statistical significance was accepted when *p* ≤ 0.05, and the tendency was considered when 0.05 < *p* ≤ 0.10.

#### 2.6.1. Behaviors in Piglets and Sows around Weaning

For each behavior category, the behavioral data on each observation day were summed up. The proportion of each behavior category was obtained by dividing the amount of the behavior of a pen by the total amount of sample points of that observation day. Each behavior category was expressed as a proportion of total active behavior.

The categories “Other social interaction” and “Object play and exploration” (piglet), and “Pen investigation” (sow) were normally distributed, so they were analyzed in linear mixed models (LMM). There were 30%, 95% and 93.3% of 0 s in Biting, Belly-nosing and Locomotor play (piglet), respectively, so Biting was merged into the category of “Aggression”; 84.4%, 68.9% and 62.2% of 0 s in Biting, Object investigation and Locomotion (sow), respectively, so Biting was merged into the category of “Aggression”. Belly-nosing and Locomotor play (piglet), and Object investigation and Locomotion (sow) were not analyzed. “Aggression + Biting” (piglet), “Aggression + Biting”, and Other social interactions (sow) were log(1 + x) transformed and analyzed in LMMs. Vocalization (sow) was analyzed in a GLMM with a Poisson distribution. In all models, the behavior was as the response variable, farrowing system, day and the interaction between farrowing system and day were the fixed effects, and the batch was the random factor.

#### 2.6.2. Bodyweight and Average Daily Gain in Piglets around Weaning

Bodyweight on 1D pre- (BW_1D pre-weaning_) and 5D post-weaning (BW_5D post-weaning_), and ADG around weaning were analyzed in LMMs: BW_1D pre-weaning_, BW_5D post-weaning_, or ADG (the means of each nursery pen) as the response variable, farrowing system as the fixed effect, bodyweight on D3 (BW_3_) as the covariate, and batch as the random factor.

#### 2.6.3. Number of Aggression-Associated Skin Lesions in Piglets around Weaning

The total number of skin lesions was summed by the individual due to a low number of lesions in each body part. The number of skin lesions was not normally distributed after square root transformation. A GLMM with a Poisson distribution was used to analyze the untransformed data: number of skin lesions (the means of each nursery pen) as the response variable, farrowing system, day, and the interaction between farrowing system and day as the fixed effects, and batch as the random factor.

#### 2.6.4. Salivary Stress Biomarkers in Piglets and Sows around Weaning

Salivary stress biomarkers, including CORT and CgA, were log-transformed and analyzed in LMMs: concentration of CORT or CgA (piglet model: the means of each nursery pen; sow model: the means of each group pen) as the response variable, farrowing system, day, and the interaction between farrowing system and day as the fixed effects, the basal level (i.e., samples on 1D pre-weaning) as the covariate, and batch as the random factor.

## 3. Results

We present the results by piglets and sows. Two JLF15 sows in the winter batch, one JLF15 sow in the spring batch, one FC sow in the summer batch, and two SWAP sows in the summer batch were culled after weaning and therefore were excluded from the analysis. Information regarding the crating period (in days), average litter size, and the reproductive parameters of the sows in each farrowing system can be found in Ko et al. [11].

### 3.1. Piglet

#### 3.1.1. Post-Weaning Behavior

The proportion of each active behavior observed in piglets after weaning is presented in Table 3. Overall, we did not find any difference between farrowing systems in the post-weaning behaviors.

Within each farrowing system, aggression and biting in SWAP piglets increased from the day of weaning to 1D post-weaning (*p* = 0.009) and the increase persisted on 2D post-weaning (*p* = 0.05, between D24 and D26), whereas that in FC and JLF15 piglets neither increased nor decreased. Other social interactions in SWAP piglets decreased from the day of weaning to 2D post-weaning (*p* = 0.03), whereas in FC and JLF15 piglets it remained similar around weaning. Object play and exploration in SWAP piglets decreased from the day of weaning to 1D post-weaning (*p* = 0.01) and then increased on 2D post-weaning (*p* = 0.03), while no changes were observed in either FC nor JLF15 piglets.

#### 3.1.2. Aggression-Associated Skin Lesions around Weaning

Table 4 presents the number of skin lesions counted in pre- and post-weaning periods in each of the three farrowing systems. No difference in the number of skin lesions was observed between farrowing systems (*p* = 0.62). Regardless of the farrowing systems, there is a trend in the number of skin lesions over time, which first increases after weaning (between 1D pre- and 1D post-weaning, all farrowing systems *p* < 0.001), and then decreases a day after (between 1D and 2D post-weaning, all farrowing systems *p* < 0.01).

#### 3.1.3. Salivary Stress Biomarkers Post-Weaning

Table 5 presents the concentration of CORT and CgA in piglets in pre- and post-weaning periods in each of the three farrowing systems. No difference in CORT and CgA was observed between farrowing systems (*p* = 0.55 and 0.58, respectively).

CORT in JLF15 increased from 1D pre- to 1D post-weaning (*p* = 0.02), and the increase persisted for 2 days (*p* = 0.01, between D23 and D26), whereas that in FC and SWAP remained at a similar level around weaning. CgA in FC increased from 1D post- to 2D post-weaning (*p* = 0.04), whereas that in SWAP and JLF15 remained at a similar level around weaning.

#### 3.1.4. Bodyweight (BW) and Average Daily Gain (ADG) around Weaning

Table 6 presents the BWs in 1D pre- and 5D post-weaning and the ADG around weaning in each of the three farrowing systems. There was no difference in BW_1D pre-weaning_, BW_5D post-weaning_, and ADG around weaning between three farrowing systems (*p* = 0.43, 0.61 and 0.16, respectively).

### 3.2. Sow

#### 3.2.1. Post-Weaning Behavior

The proportion of each active behavior observed in sows after weaning in each of the three farrowing systems is presented in Table 7. Except for vocalization (*p* < 0.0001), there was no difference between farrowing systems in other post-weaning behaviors. The trend over time of the vocalization counts after weaning was similar in all farrowing systems: Sows vocalized the most on the day of weaning, and then rapidly decreased on 1D post- and 2D post-weaning (all farrowing systems, *p* ≤ 0.0001, between D24 and D25, and between D24 and D26).

#### 3.2.2. Salivary Stress Biomarkers Post-Weaning

Table 8 presents the concentration of CORT and CgA in sows in pre- and post-weaning periods in each of the three farrowing systems. No difference in CORT and CgA was observed between farrowing systems (*p* = 0.81 and 0.22, respectively).

## 4. Discussion

This paper studies the effect of farrowing systems (i.e., early environment) on the adaptability to weaning in sows and piglets on a commercial farm. It is a follow-up study from Ko et al. [11] of the post-weaning period, meaning the same group of animals from Ko et al. [11] were followed. Sows and piglets were no longer housed in litters but regrouped in weaning facilities. As the first published pilot study involving farrowing pens with temporary crating in Spain, it is interesting to follow these sows and piglets from lactation to a few days after weaning, and how they adapted to weaning challenges in commercial conditions.

### 4.1. Piglets

Similar numbers of skin lesions after weaning between farrowing systems indicated that the level of aggression within a group after regrouping was similar, regardless of farrowing systems. The result agrees with Chaloupková et al. [14] and Verdon et al. [21], where they found that piglets raised in FC were as aggressive (e.g., frequency of aggression and fights, fighting duration, and number of skin lesions) as those raised in farrowing pens (i.e., without sow confinement throughout the lactation period) but a single litter after weaning. As stated in Chaloupková et al. [14], a series of stressful events at weaning, including the change of the environment, maternal separation, and mixing with unfamiliar individuals, may mask the differences of the pre-weaning housing environment. Providing a socially enriched pre-weaning environment, such as early exposure to non-littermates, may play a more important role to reduce aggression after regrouping, especially at weaning [7,8,14,22].

We hypothesized more frequent mother–young interactions in TC during lactation may lead to an additional increased weaning stress response in sows and piglets, due to the separation of a closer sow-piglet bonding compared to FC [16]. This was, however, not the case in sows or piglets. SWAP piglets showed a similar level of CORT and CgA post-weaning, suggesting that weaning, known as a stressful event in pig production [23], did not activate the HPA (Hypothalamic-Pituitary-Adrenal) and the SAM (Sympathetic-Adrenal-Medullary) axes in SWAP piglets. We did not expect to see the difference in salivary stress biomarkers between SWAP and JLF15 as both systems are TC. However, some of the differences in pen features between SWAP and JLF15 may explain it. Firstly, SWAP pens have a slightly larger (1.04-times larger) space allowance than JLF15. Secondly, a larger coverage of the solid flooring in SWAP pens can retain the nest-building materials (hay in this case) on the floor for piglets to manipulate during the suckling period. Thirdly, a metal-barred gate is installed between the adjacent SWAP pens, allowing the piglets to perform nose contact with non-littermates and the neighbor sow before weaning (Figure 2). Lastly, the fixed-side crate of SWAP pens does not block the sows’ direct contact with the creep area (Figure 3). All in all, an insignificant weaning stress response in SWAP piglets may be due to the additional facilitation of exploration and social behaviors in a larger and a more enriched (both physically and socially) early environment, which is known to mitigate the weaning stress (physical enrichment: environmental enrichment [10], space allowance [24]; social enrichment: nose contacts [25], early social play [26], early socialization [8]). However, we acknowledge that in Ko et al. [11], we did not find a difference in social interactions between piglets (defined as SB in the paper), exploration in piglets (defined as PPE), and piglet-initiated mother–young interactions (defined as NNC and SC) between SWAP and JLF15 during the suckling period. Better adaptability to weaning in SWAP piglets, therefore, deserves a deeper understanding by different approaches, and certainly more studies comparing the welfare and performance in sows and piglets in different TC are needed.

### 4.2. Sows

According to the farmer’s observations, sows on this farm commonly produce high-pitched calls after weaning for a few days when they were moved from the farrowing pens to the group pens. In the present study, we found that SWAP sows vocalized more than FC and JLF15 sows on the day of weaning and a day after. Moreover, a similar trend in the change of vocalization counts was found across farrowing systems, which peaked on the day of weaning and rapidly decreased over time (with a 5-time reduction in FC and JLF15, and a 15-time reduction in SWAP, from the day of weaning to 2-day post-weaning). As Manteuffel et al. [27] reviewed, louder, longer, and high-pitched calls (e.g., squeals and screams) in pigs are related to the state of the excitement. Additionally, Xin et al. [28] found that high frequency and long duration of calls can imply more severe stress in pigs, although it was not physiologically validated. It is known that a close bonding between the sow and the piglets is established already in early lactation, with piglets recognizing the odor and the vocalization from their mother by 3 days of age [29]. In Ko et al. [11], we observed that more mother–young interactions were found in SWAP and JLF15 than FC during lactation, including both sow- and piglet-initiated. We therefore suspect the relation between this high-pitched call with the separation from the piglets, although we acknowledge that SWAP sows vocalized significantly more than JLF15. However, to the best of our knowledge, we did not find studies about vocalization in sows related to separation from the piglets or weaning, so it is difficult to demonstrate the association. Yet, further investigation on the high-pitched call produced after weaning by sows is warranted to confirm the suspicion.

A similar level of salivary CORT and CgA in sows around weaning indicated an insignificant activation of the HAP and SAM stress axes. This could be due to several reasons that may overshadow the stress of separation from the piglets. Having a dominant boar [30] in the adjacent pen helps to minimize aggression and salivary stress at regrouping, as suggested in many studies [31,32,33]. Regardless of farrowing systems, sows from the same batch were kept in the same gestation pen before being assigned to different farrowing systems, which would not induce much aggression at regrouping after weaning as they already recognized each other [30,34]. It is also true that the regrouping procedure varied in different batches due to different availability of the group pens, for instance, the stocking density and the distribution of the boars in the adjacent pens. Further research to confirm the stronger mother–young bonding during lactation and its impact on the separation after weaning in loose pen sows and piglets is therefore necessary.

### 4.3. Limitations of the Present Study

We acknowledge that there are two main limitations of the present study: low sample size and commercial settings. A low sample size gives a relatively weak statistical power to interpret the results. However, using farrowing pens with temporary crating is becoming a trend in the European Union, and studies regarding this topic are rather scarce. We therefore would like to encourage more research on the effect of this system, especially on the weaning adaptability in piglets. In addition, conducting studies in commercial farms where the control over different variables is rather limited (availability of the housing facilities after weaning which led to different regrouping procedures in different batches, in this case). The results of the present study ought to be interpreted with caution.

## 5. Conclusions

Based on the results collected from this pilot study, farrowing pens with temporary crating which allow more mother–young interactions, did not seem to reduce aggression in piglets after weaning. However, salivary stress biomarkers showed a lower weaning stress response in SWAP than FC and JLF15 piglets. Increased high-pitched calls in SWAP sows after weaning might be associated with the stress of the separation from the piglets, but more research in sow vocalization after weaning should be studied. Due to the growing popularity of using alternative farrowing systems, more research to study the effect of the farrowing/early environment, which promotes close contact between the mother and the young, on a pig’s life-long adaptability towards social challenges (e.g., weaning or regrouping) is needed.

## Figures and Tables

**Figure 1 animals-12-00724-f001:**
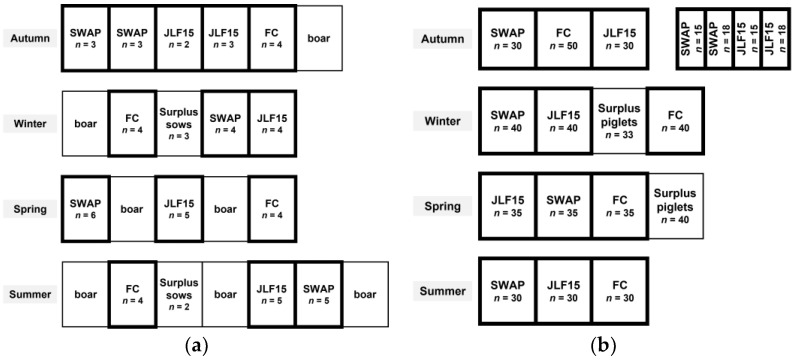
Distribution of (**a**) sows and (**b**) piglets after weaning in four batches, including 14 group pens for the sows and 16 nursery pens for the piglets, which are highlighted in thick lines. (FC: pigs that were housed in the conventional farrowing pens with a farrowing crate before weaning; SWAP (Sow Welfare and Piglet protection) and JLF15: pigs that were housed in two types of farrowing pens with temporary crating before weaning.) *n* indicates the number of the sows or piglets in one pen. A pen that has sows/piglets originating from more than one farrowing system is considered as the “surplus sows/piglets”, and they are discarded from the study after weaning.5.

**Figure 2 animals-12-00724-f002:**
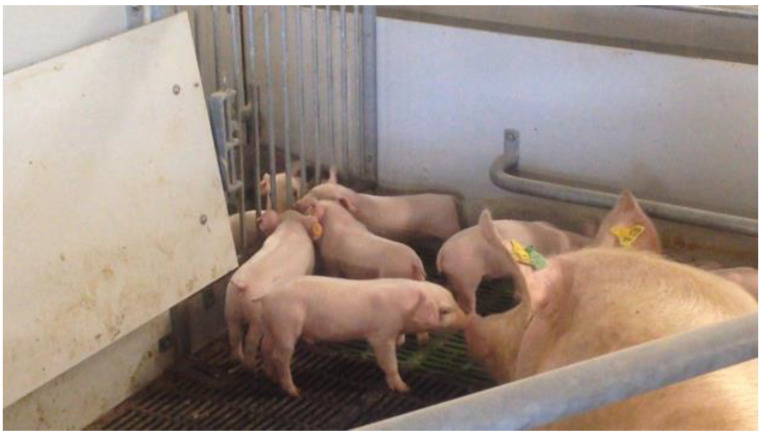
A metal-barred gate installed between the adjacent SWAP pens.

**Figure 3 animals-12-00724-f003:**
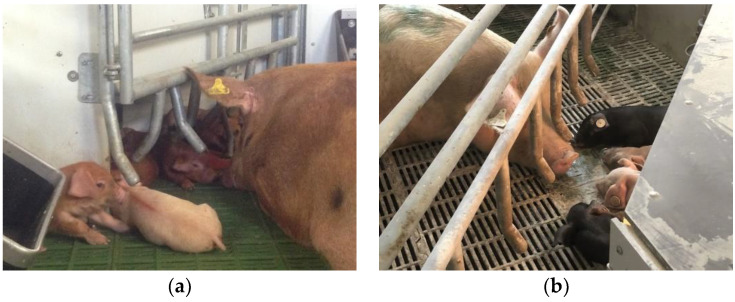
Different distances for the sow to access the creep area in (**a**) SWAP and (**b**) JLF15 pens.

**Table 1 animals-12-00724-t001:** Behavior categories for post-weaning piglets.

Behavior	Description
Aggression	Any physical interaction that indicates social conflict with brute force involving two or more individuals, including head knocking, pushing and fighting. The number of the fighting event is recorded according to the number of individuals involved (adapted from Ko et al. [8]).
Biting	Piglet orally in contact with another piglet, including nibbling, chewing, sucking and biting. Biting during fighting is excluded (adapted from Day et al. [17]).
Belly-nosing	Repetitively nosing with a rhythmic pattern towards another piglet’s abdomen with its snout (adapted from Li and Gonyou [18]).
Other social interaction	Any physical interaction with minimal or moderate force involving two or more individuals with no reaction or avoidance response from the recipient(s), including sniffing, nudging, mounting and chasing (adapted from Ko et al. [8]).
Locomotor play behavior	See Martin et al. [9] for different locomotor play behaviors.
Object play and exploratory behavior	Behaviors including sniffing or manipulating pen features, enrichment objects or other items; feed and water are excluded.

**Table 2 animals-12-00724-t002:** Behavior categories for post-weaning sows.

Behavior	Description
Aggression	See the definition of Aggression in Table 1.
Biting	Sow orally in contact with another animal, including nibbling, chewing and biting. Biting during fighting is excluded.
Other social interaction	See the definition of Other social interaction in Table 1. Oral contact is excluded.
Pen investigation	Sow sniffing or manipulating pen features (adapted from Averós et al. [19]).
Object investigation	Sow sniffing or manipulating the enrichment object (adapted from Averós et al. [19]).
Vocalization	Sow produces a certain type of high-pitched call. Normal grunting, vocalization during fighting or being bitten is excluded.
Locomotion	Any locomotor behavior not listed above, for example, walking and pacing.

**Table 3 animals-12-00724-t003:** Proportion (number of behavior/total active behaviors) of active behaviors in piglets (mean ± SE) in the post-weaning period in three farrowing systems (FC, *n* = 4: conventional farrowing crate; SWAP, *n* = 6, and JLF15, *n* = 6: farrowing pens with temporary crating). Values with a different superscript (^a^, ^b^) correspond to a significant difference (*p* ≤ 0.05) between observation days in the same behavior category and farrowing system. *n* indicates the number of nursery pens.

	Aggression and Biting ^1^	Other Social Interaction	Object Play and Exploration
Day of weaning (D24)			
FC	0.07 ± 0.02	0.44 ± 0.09	0.49 ± 0.09
SWAP	0.06 ± 0.01 ^a^	0.37 ± 0.04 ^a^	0.57 ± 0.03 ^a^
JLF15	0.10 ± 0.04	0.34 ± 0.08	0.55 ± 0.06
1D post-weaning (D25)			
FC	0.24 ± 0.09	0.41 ± 0.08	0.34 ± 0.08
SWAP	0.28 ± 0.07 ^b^	0.32 ± 0.05 ^a,b^	0.36 ± 0.07 ^b^
JLF15	0.27 ± 0.08	0.26 ± 0.05	0.47 ± 0.07
2D post-weaning (D26)			
FC	0.23 ± 0.05	0.28 ± 0.05	0.48 ± 0.06
SWAP	0.22 ± 0.05 ^b^	0.22 ± 0.05 ^b^	0.55 ± 0.09 ^a^
JLF15	0.18 ± 0.04	0.24 ± 0.05	0.57 ± 0.07
Global *p*-value	0.99	0.32	0.40

^1^ Aggression and Biting are presented as back-transformed values.

**Table 4 animals-12-00724-t004:** Number of skin lesions of piglets (mean ± SE) counted on D23 (1D pre-weaning), D25 (1D post-weaning) and D26 (2D post-weaning) in three farrowing systems (FC: conventional farrowing crate; SWAP and JLF15: farrowing pens with temporary crating). Values with a different superscript (^a^, ^b^) correspond to a significant difference (*p* ≤ 0.05) between sampling days in the same farrowing system. *n* indicates the number of nursery pens.

Farrowing System	*n*	1D Pre-Weaning	1D Post-Weaning	2D Post-Weaning
FC	4	0.74 ± 0.20 ^a^	7.31 ± 2.87 ^b^	4.93 ± 2.38 ^b^
SWAP	6	1.02 ± 0.40 ^a^	7.77 ± 1.96 ^b^	5.77 ± 0.93 ^b^
JLF15	6	0.96 ± 0.23 ^a^	9.11 ± 2.54 ^b^	6.69 ± 1.25 ^b^

**Table 5 animals-12-00724-t005:** Concentration of salivary cortisol (CORT) and chromogranin A (CgA) of piglets (mean ± SE) collected on D23 (1D pre-weaning), D25 (1D post-weaning) and D26 (2D post-weaning) in three farrowing systems (FC: conventional farrowing crate; SWAP and JLF15: farrowing pens with temporary crating). Values with a different superscript (^a^, ^b^) correspond to a significant difference (*p* ≤ 0.05) between sampling days in the same farrowing system. *n* indicates the number of nursery pens.

Farrowing System	*n*	1D Pre-Weaning	1D Post-Weaning	2D Post-Weaning
CORT, µg/dL				
FC	4	1.01 ± 0.16	1.42 ± 0.46	1.20 ± 0.27
SWAP	6	0.84 ± 0.17	1.34 ± 0.46	1.56 ± 0.45
JLF15	6	0.75 ± 0.11 ^a^	1.78 ± 0.39 ^b^	1.83 ± 0.39 ^b^
CgA, µg/mL				
FC	4	0.51 ± 0.09 ^a^^,^^b^	0.50 ± 0.20 ^b^	0.76 ± 0.26 ^a^
SWAP	6	0.71 ± 0.17	0.71 ± 0.22	0.57 ± 0.17
JLF15	6	0.50 ± 0.14	0.58 ± 0.14	0.61 ± 0.09

**Table 6 animals-12-00724-t006:** Bodyweight (BW) (kg) on D23 (1D pre-weaning) and D29 (5D post-weaning), and average daily gain (ADG) (g/day) between D23 and D29 in piglets in three farrowing systems (FC: conventional farrowing crate; SWAP and JLF15: farrowing pens with temporary crating). *n* indicates the number of nursery pens.

	Farrowing System	*n*	Mean	SEM	*p*-Value
BW_D23 (1D pre-weaning)_, kg	FC	4	5.77	0.34	0.43
	SWAP	6	5.65	0.43	
	JLF15	6	6.13	0.40	
BW_D29 (5D post-weaning)_, kg	FC	4	6.42	0.45	0.61
	SWAP	6	6.14	0.37	
	JLF15	6	6.68	0.40	
ADG_D23-29_, g/day	FC	4	108.00	20.73	0.16
	SWAP	6	81.70	14.09	
	JLF15	6	91.69	12.97	

**Table 7 animals-12-00724-t007:** Proportion (number of behavior/total active behaviors) of active behaviors in sows (mean ± SE) in the post-weaning period in three farrowing systems (FC, *n* = 4: conventional farrowing crate; SWAP, *n* = 5, and JLF15, *n* = 4: farrowing pens with temporary crating). Values with a different superscript (^a^, ^b^, ^c^) correspond to a significant difference (*p* < 0.05) between sampling days in the same farrowing system. Values with a different superscript (^x^, ^y^) correspond to a significant difference (*p* ≤ 0.05) between farrowing systems on the same sampling day. *n* indicates the number of pens.

	Aggression and Biting ^1^	Other Social Interaction	Pen Investigation	Vocalization (Average Counts)
Day of weaning (D24)				
FC	0.03 ± 0.01	0.22 ± 0.08	0.42 ± 0.04	25.50 ± 9.54 ^a,x^
SWAP	0.12 ± 0.04	0.24 ± 0.06	0.34 ± 0.07	52.80 ± 6.45 ^a,y^
JLF15	0.17 ± 0.05	0.37 ± 0.08	0.32 ± 0.06	19.25 ± 7.58 ^a,x^
1D post-weaning (D25)				
FC	0.13 ± 0.11	0.23 ± 0.10	0.35 ± 0.14	6.00 ± 3.34 ^b,x^
SWAP	0.05 ± 0.03	0.24 ± 0.11	0.46 ± 0.14	19.80 ± 8.21 ^b,y^
JLF15	0.16 ± 0.03	0.30 ± 0.17	0.43 ± 0.15	7.75 ± 6.12 ^b,x^
2D post-weaning (D26)				
FC	0.12 ± 0.04	0.24 ± 0.07	0.33 ± 0.14	5.25 ± 5.25 ^b^
SWAP	0.10 ± 0.05	0.31 ± 0.11	0.27 ± 0.09	3.60 ± 3.60 ^c^
JLF15	0.07 ± 0.06	0.19 ± 0.12	0.49 ± 0.22	3.75 ± 3.75 ^c^
Global *p*-value	0.64	0.97	0.88	<0.0001

^1^ Aggression and Biting are presented as back-transformed values.

**Table 8 animals-12-00724-t008:** Concentration of salivary cortisol (CORT) and chromogranin A (CgA) of sows (mean ± SE) collected on D23 (1D pre-weaning), D25 (1D post-weaning) and D26 (2D post-weaning) in three farrowing systems (FC: conventional farrowing crate; SWAP and JLF15: farrowing pens with temporary crating). *n* indicates the number of group pens.

Farrowing System	*n*	1D Pre-Weaning	1D Post-Weaning	2D Post-Weaning
CORT, µg/dL				
FC	4	1.49 ± 1.08	1.24 ± 0.41	1.49 ± 0.39
SWAP	4	0.89 ± 0.19	1.42 ± 0.46	1.72 ± 0.52
JLF15	4	0.67 ± 0.12	1.28 ± 0.49	1.65 ± 0.65
CgA, µg/mL				
FC	4	0.57 ± 0.15	1.22 ± 0.52	1.10 ± 0.40
SWAP	4	0.81 ± 0.44	1.11 ± 0.56	1.29 ± 0.41
JLF15	4	0.94 ± 0.31	1.39 ± 0.51	0.66 ± 0.21

## Data Availability

The data presented in this study are available on reasonable request from the corresponding author.

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
