# Peer review of "Welfare and Performance of Post-Weaning Sows and Piglets Previously Housed in Farrowing Pens with Temporary Crating on a Spanish Commercial Farm: A Pilot Study"

_animals, 2022, doi:10.3390/ani12060724_

Round 1

Reviewer 1 Report

The manuscript addresses a current area of research interest, particularly the housing in the farrowing house. The manuscript is well written, the M&M is well described. I have just 2 small suggestions: 

Line 125: I miss the breed/hybrid of the sows and the piglets in '2.2. Animals and management'.

Table 1 + 2: Please cite the references on which the ethograms (for sows and piglets) are based.  

Author Response

Thank you very much for your feedback.

We have added the breed of the pigs into the manuscript (Line 92-93) and also some references for the ethograms (please see Table 1 and 2).

Reviewer 2 Report

The paper deals with a topic of great practical and scientific interest, namely the development of rearing techniques aimed at minimizing the stress associated with weaning of piglets.

The scientific hypothesis is clearly outlined and adequately supported by the review of the existing literature on the subject. The authors correctly highlight the limitations of the present study, due to the low sample size and commercial settings. I add another critical issue that may have undermined the overall validity of the results obtained: the piglets were weighed on the same day in which the saliva samples were taken for the cortisol and CgA measure. The time at which the weighing of the piglets was carried out and the procedure used are not specified. However, if, as usual, the weighing was carried out in the early morning and this involved the handling of the piglets, it is conceivable that this has distorted the values of the stress biomarkers measured at D23. This circumstance must therefore be added to the limitations of the present study, together with the slightly different stocking density between the groups of animals in the experiment.

Beyond these criticisms, the experimental design is well structured, the analytical determinations are congruous and the statistical analysis of the data is appropriate. However, it is advisable not to refer to the previous work by Ko et al. for figures, tables and procedures, since the work must be self-consisting. Both the figure and the table must be presented in the body of this work and the procedure for saliva collection can be briefly recalled.

It is also advisable, for greater ease of reading, that the statistical differences existing not only within each treatment, but also between the experimental treatments, are highlighted in all the tables with different superscripts.

With the limitations highlighted, the description of the results and discussion of them appears appropriate. The list of references is wide, adequate and updated

Author Response

Thank you very much for your feedback.

Thank you for pointing out that on D23, both saliva sampling and weighing were carried out. We first took the saliva samples and then weighed the piglets to avoid the confounding effect of the handling stress. We have added this sentence in M&M to clarify this concern (Line 180-181).

To avoid referring to the previous paper repeatedly, we have added some sentences to briefly describe the management of the animals (Line 89-92) and the lab procedures for salivary analysis (Line 173-175).

As for the highlight of the statistical difference between the experimental treatments in tables, we did present it when there is a statistical difference (like in Table 7). In most of the tables, there are no difference between the experimental treatments, so we did not present the superscripts.